

# Io's auroral emissions via global hybrid plasma simulations

Štěpán Štverák[1,2], Pavel Trávníček[3], Ondřej Šebek[1], and David Herčík[1]

[1]Institute of Atmospheric Physics, CAS, Prague, Czech Republic.
[2]Astronomical Institute, CAS, Ondřejov, Czech Republic.
[3]Space Sciences Laboratory, UCB, Berkeley, CA, USA.

**Correspondence:** Štěpán Štverák (stverak@ufa.cas.cz)

**Abstract.** We tackle the Io's aurora source and topology by carrying out a set of global hybrid simulations of Io's interaction with the plasma torus under different model geometry and background conditions. Based on the simulated results, we compute the photon emission rates above the Io's surface and present the resulting images from a virtual telescope and topological maps showing the distribution of the emission sources across the moon's surface. This allows us to compare the structure of

the aurora with the real observations and conclude on the different assumptions. We found a reasonable agreement with the real observations in the case of non-collisional background electron populations. From the comparison of the local magnetic field topology with model aurora structures, we also infer that an induced dipole feature is more probable to play a role in the interaction of Io with the Jovian magnetosphere. In addition we also examine the potential contribution of energetic electron beams, being observed in the Io's wake region, to the overall auroral emissions.

## 1   Introduction

The volcanism of Io is a significant matter contributor to the Jupiter's magnetosphere. Its strong volcanism supplies the moon's atmosphere with volcanic material which makes the sulphur and oxygen neutrals the dominant species. These neutrals get, eventually, ionized and then these atmospheric particles supply Jupiter's magnetosphere with plasma, creating dense plasma torus on the Io's orbit. This plasma torus interacts with Jupiter's magnetospheric magnetic field and the interactions cause

various features and effects within the Jupiter's magnetosphere and in the Jupiter's atmosphere itself. One such phenomenon, which has been subject of several studies, are enhanced auroral emissions within Jupiter's ionosphere around the footprint of the connected Io's flux tube (Connerney et al., 1993). The interaction of Io's atmosphere with the torus plasma formed by the structure of Jupiter's magnetic field, possibly also superposed with induced dipole moment of Io (Khurana et al., 2011), cause, however, ionization and excitation within the Io's own atmosphere resulting in aurora emissions observed on Io (Roesler et al.,

20   1999).

    The atmosphere of Io may be supplied by material either directly from Io's volcanic eruptions (volcanism driven atmosphere) and/or from sublimation of the surface frosts (sublimation driven atmosphere). Several researchers have addressed the question which of these two sources dominates and obtained different results. For instance Moses et al. (2002) demonstrated that volcanism and sublimation driven atmospheres would exhibit different composition and suggested that observed high abundance

of SO indicates the importance of volcanic supply of Io's atmosphere. Saur and Strobel (2004) developed a model of Io's



auroral response to changes in the neutral atmosphere during and after eclipse. Their results indicate that contribution from frost sublimation exceeds the contribution from volcanoes by a factor of $\approx 10$ in sunlight. Jessup et al. (2004) observed a falloff of $SO_2$ density away from sub-solar point and suggested that this indicates an atmosphere in vapor pressure equilibrium with surface frost. Geissler et al. (2004) analyzed Cassini's observations of Io's aurora and showed that the auroral features persist

during eclipse suggesting that the atmosphere is substantially supported by volcanism. On the other hand New Horizons' observations of aurora during eclipse analyzed by Retherford et al. (2007) indicate that volcanoes supply $1-3\%$ of the day-side atmosphere. Numerical modeling of the atmosphere performed by Walker et al. (2012) indicates that the atmosphere may be supplied purely by the frost sublimation except for regions close to active volcanoes. Tsang et al. (2012, 2013b) investigated correlations between properties of anti-Jovian atmosphere and distance from Sun and concluded that the supplies from volca-

noes and sublimation are comparable but the sublimation driven supply slightly dominates. Later Tsang et al. (2015) showed that the sub-Jovian atmosphere may be dominated by the volcanic supply.

The neutral atmosphere consists mainly of sulfur dioxide, $SO_2$, first detected by Pearl et al. (1979). Sulfur monoxide was detected by Lellouch et al. (1996) and its day side mixing ratio is assumed to be below $10\%$ (e.g., Lellouch et al., 1996; Wong and Johnson, 1996; Wong and Smyth, 2000; Jessup et al., 2004; Tsang et al., 2013a). Atomic oxygen and sulfur were also

found at Io (e.g., Oliversen et al., 2001; Wolven et al., 2001, and references therein) and their mixing ratios are assumed to be of the order of percents. Other atomic and molecular components such as sodium (Brown, 1974), $S_2$ (Spencer et al., 2000), NaCl (Lellouch et al., 2003) or KCl (Moullet et al., 2013) are present to a lesser extent. Estimates of $SO_2$ column density vary from $\approx 10^{16}$ $cm^{-2}$ to $\approx 10^{18}$ $cm^{-2}$ and estimates of atmospheric temperature vary in range $\approx 100-500$ K (see Walker et al., 2010; Tsang et al., 2012, and references therein).

Particles in Io's neutral atmosphere may be excited in various processes and subsequent emission of photons produces Io's aurora. Variety of discrete spectral lines corresponding to different species and excited states may exist. Usually emissions from neutral oxygen and sulfur are observed, the typical emissions are OI1304, OI1356, SI1479, SI1900, here a standard notation is used where the Roman number identifies charge state of the atom (I for neutral, II for once ionized, etc.) and the Arabic number defines wavelength of emitted photon in ångströms (Å). There is a long history of observing Io in various parts of

spectra, however, the early observations allowed only for disk-integrated spectra and served mainly as a tool for identifying species in Io's environment (Brown, 1974; Lellouch et al., 1992, 1996, e.g.,).

First spatially resolved observations of Io at far-ultraviolet wave-lengths by the Hubble Space Telescope (Roesler et al., 1999) revealed several distinct features in Io's aurora, namely bright equatorial spots close to sub-Jovian and anti-Jovian points, limb glow at high latitudes of Io, and so-called extended emission associated with Io's wake. Roesler et al. (1999)

found that latitude of both equatorial spots is strongly correlated with the direction of background magnetic field. Both spots shift from the equator towards northern or southern latitude (one spot to each side) in such way that a line connecting both spots is approximately perpendicular to the background magnetic field. Moreover Roesler et al. (1999) found that brightness of limb glows is correlated with position of Io in the plasma torus so that the limb glow emission from the hemisphere facing torus centrifugal equator is brighter than that on the other hemisphere. Retherford et al. (2000) performed detailed analysis of

the far-ultraviolet auroral emissions of Io and specified the previous findings, most notably they inferred that the equatorial





spots are shifted somewhat downstream from the sub-Jovian/anti-Jovian meridians and that the spot on anti-Jovian side of Io is brighter than the sub-Jovian equatorial spot. Retherford et al. (2000) also found a correlation between brightness of equatorial spots and distance of Io from torus centrifugal equator. This correlation can be most easily attributed to the change of electron density with position of Io in the plasma torus. Later Retherford et al. (2003) analyzed in detail behavior of the limb glow

emissions and concluded that their brightness is governed by amount of electron energy contained in the flux tube connected to given hemisphere of Io. Most of the features of far-ultraviolet aurorae were observed also in the visible spectra (Geissler et al., 1999, 2001). In contrast to the far-ultraviolet observations the visible auroral emissions exhibit correlation in brightness with proximity to active volcanic plumes. Geissler et al. (2001) ascribe this correlation to enhanced concentration of sulfur dioxide molecules originating from the plumes.

Above mentioned observations indicate that the aurora of Io is a dynamic phenomenon which reacts flexibly to changing interaction conditions and also reflects local properties in the system. Since most of the emissions are assumed to result from excitation by electrons (mainly thermal) the auroral structure may reveal information about pattern of electron flow around Io, electron temperature, etc. There were efforts to explain observed auroral morphology in the context of numerical modeling or theoretical analysis. Saur et al. (2000) modeled auroral morphology on the basis of two-fluid plasma model. They showed that

the electron fluid loses energy in excitation processes preferentially on the upstream side of Io and on its flanks. Collection of local emission rate along the flow direction then naturally produces bright spots at the flanks of Io where the line of sight goes through longest region of emission. According to their model the observed asymmetry in brightness of equatorial plasma results from asymmetry of the electron flow pattern which is introduced by the Hall effect. Due to this effect the electron flow is unequally diverted around Io so that higher fraction of the upstream electrons flows past Io along the anti-Jovian flank.

Observations indicate that positions of the equatorial spots correspond to positions where the background Jovian magnetic field is tangent to Io's surface. Roth et al. (2017) used MHD model to derive magnetic field topology around Io and calculated locations of tangent magnetic field. They showed that magnetic field without contribution of induced dipole field corresponds better to assumed locations of auroral equatorial spots.

In the present paper we test some of the above hypothesis dealing with the Io's auroral morphology by use of global numerical

modeling of Io's interaction with the Jovian plasma torus. Our model is based on the hybrid approach where plasma kinetic effects are resolved down to ion scales while electrons are treated as a mass-less fluid which maintains the local quasi-neutrality. Our aim is not to exactly reproduce any of the real individual observations as taken by many ground based or space born telescopes or cameras but rather investigate the effect of various background plasma conditions on the resulting auroral patters. This in turn may provide useful clues to analyze and describe the real plasma processes and conditions near Io using remote

auroras observations as indirect measurements. The numerical simulations and models of auroral emissions used for this study are described in section 2. Section 3 then presents the obtained results either in a form of virtual telescope observations or as surface maps of the emission rates. The results with respect to previous findings are then discussed in section 4, and finally concluded in section 5.



## 2 Model

In our study we use results from numerical simulations of Io's interaction with the magnetospheric plasma presented by Šebek et al. (2019). The local interaction of Io with the background plasma torus is an example of sub-Alfvénic interaction of flowing plasma with a non-magnetized body possessing a neutral atmosphere which serves as a mass-loading source feeding the torus with cold pick-up ions through many ionization processes. Our model employs 3-D hybrid code based on the current advance method and a cyclic leapfrog algorithm developed by Matthews (1994). In the hybrid approach, ions are treated by using a particle-in-cell scheme while electrons are represented by a massless, isothermal, charge neutralizing fluid. The code self-consistently solves equations of motion for ions together with Faraday's law for the magnetic field and a generalized Ohm's law for the electric field. The model further implements multiple ion species resolving both the thermal components of the background plasma torus and the new born pick-up ions from Io's neutrals. For complete description of the numerical model and the simulation setups see Šebek et al. (2019).

| Run ID | Reference name in Šebek et al. (2019) | Description |
|---|---|---|
| *I0-C* | I0-a | Electron temperature coupled via Coulomb collisions |
| | | Induced magnetic dipole |
| *I0-NC* | I0-b | Collisionless electrons - not thermally coupled |
| | | Induced magnetic dipole |
| *I0-NC-ND* | I0-c | Collisionless electrons - not thermally coupled |
| | | No induced mognetic dipole |

**Table 1.** List of simulation runs used for the present study according to Šebek et al. (2019). All three simulation runs correspond to the Galileo flyby I0: run I0-C implements thermal coupling between background and pickup electrons via Coulomb collisions; run I0-NC keeps both background and pickup electron temperatures constant, i.e., equal to initial conditions; run I0-NC-ND is same as I0-NC wrt. electron temperatures but does not implement any induced magnetic field of Io.

In particular, we have selected three simulation runs corresponding to the Galileo flyby I0, see Table 1. All three simulation runs have equal initial plasma conditions as given in Table 2. Simulations *I0-C* and *I0-NC* further include Io's induced magnetic dipole field with an equatorial magnitude of $B_{ind} = 300\,nT$ and the magnetic moment being anti parallel to the background magnetic field. The common coordinate system has the $+X$ axis aligned with the background (torus) plasma flow, the $+Y$ axis oriented radially toward Jupiter, and the $Z$ axis completes a right-handed coordinate system, pointing mostly in the Jupiter's magnetic field direction in the vicinity of Io.

The model of the neutral atmosphere, representing the source of pickup ions, is composed of atomic oxygen, and sulfur; and molecular sulfur oxide and sulfur dioxide. A generic model of the atmosphere is used similar to previous modeling of the Io plasma interaction (e.g. Dols, 2012; Roth et al., 2011; Saur et al., 2002). The atmosphere consists of a dense lower atmosphere rapidly vanishing with distance and an extended neutral cloud. The atmosphere is symmetric in longitude with most of the





content concentrated around Io's equatorial plane (Feldman et al., 2000; Roesler et al., 1999). Each component of the neutral atmosphere model reads as follows:

$$n_{\mathrm{n}}(r,\vartheta) = n_{\mathrm{atm}} \left\{ W_{\mathrm{int}} \exp\left[ -\frac{r - R_{\mathrm{Io}}}{H_{\mathrm{atm}}} \right] + W_{\mathrm{ext}} \left( \frac{R_{\mathrm{Io}}}{r} \right)^{3.5} \right\} \cdot \left\{ W_{\mathrm{pole}} + W_{\mathrm{eq}} \exp\left[ -\left( \frac{\vartheta}{0.625} \right)^{6} \right] \right\}, \qquad (1)$$

where $r$ is distance from Io's center given in Io radii $R_{Io}$, $\vartheta \in <-\frac{\pi}{2}, \frac{\pi}{2}>$ is the latitude, $n_{\mathrm{atm}}$ is surface neutral density, $W_{\mathrm{int}}$, $W_{\mathrm{ext}}$ are relative fractions of Io's inner atmosphere and extended neutral clouds ($W_{\mathrm{int}} + W_{\mathrm{out}} = 1$), $H_{\mathrm{atm}}$ is the scale height

of the inner atmosphere, and $W_{\mathrm{pole}}$ and $W_{\mathrm{eq}}$ are relative fractions of the latitudinally symmetric/asymmetric components of the atmosphere ($W_{\mathrm{pole}} + W_{\mathrm{eq}} = 1$). Values of individual parameters used for selected simulation runs are given in Table 3.

| Initial plasma conditions | |
| --- | --- |
| Background magnetic field $B_0$ | $1840\ nT$ |
| $\mathbf{B}/B_0 = (B_x, B_y, B_z)$ | $(-0.1549, -0.0516, -0.987)$ |
| Background electron density $n_{e0}$ | $3500\ cm^{-3}$ |
| Background electron temperature $T_{e0}$ | $5\ eV$ |
| Background ion densities $O^+, S^+, S^{++}$ | $n_{O+} = 0.36n_{e0},\ n_{S+} = 0.08n_{e0},\ n_{S++} = 0.28n_{e0}$ |
| Background ion temperature $T_{i0}$ | $100\ eV$ |
| Plasma flow velocity $u_0$ | $57\ km/s$ |
| Pickup electron/ion temperature $T_{pu}$ | $0\ eV$ |

Table 2. List of initial plasma conditions common for all simulation runs.

We use our simulated results to infer morphology of Io's aurora for different interaction conditions. The main source of the auroral emissions is the excitation of neutrals by electrons. For this purpose we calculate local emission rate of photons of particular wavelength. We select a representative case of neutral oxygen emissions at wavelength of 1304 Å (hereafter referred

to as OI1304) as one of the main sources of the photo-emission observed at Io. The auroral photo emission OI1304 is within the study considered to depend on the direct electron impact excitation of oxygen and dissociative excitation of sulfur dioxide. For the excitation, two mechanisms are considered: by thermal electrons and by energetic electron beams in the Io's wake. For the calculation of electron impact excitation reactions we make similar assumptions as for electron impact ionization in Šebek et al. (2019), i.e., the production rate of photons per unit time in a given volume $\Delta V$ is given as

$$N_{\mathrm{ei}} = n_{\mathrm{n}} n_{\mathrm{e}} \nu_{\mathrm{ei}}(T_e) \Delta V. \qquad (2)$$

Here $n_{\mathrm{n}}$ is local neutral density (either O or SO$_2$), $n_{\mathrm{e}}$ is local electron density, and $\nu_{\mathrm{ei}}$ is electron impact excitation rate being a function of the local electron temperature $T_e$. The excitation rate is calculated as the mean value of electron excitation cross-section multiplied by the velocity of impacting electron from the actual electron velocity distribution function as

$$\nu_{\mathrm{ei}} = \langle \sigma_{\mathrm{ei}}(v) v \rangle = \frac{\int \sigma_{\mathrm{ei}}(v) v f_{\mathrm{e}}(\boldsymbol{v}) \mathrm{d}^3 \boldsymbol{v}}{n_{\mathrm{e}}}, \qquad (3)$$





where the velocity distribution function $f_e(\boldsymbol{v})$ is considered as simple Maxwellian for the thermal electrons or delta function for the electron beams, and the effective cross-sections $\sigma_{\mathrm{ei}}$ for the excitation reactions are taken from papers of Vaughan and Doering (1987); Gulcicek and Doering (1988) (direct excitation) and Vatti Palle et al. (2004) (sulfur dioxide dissociative excitation). The density of the neutral atmosphere $n_n$ is given by (1) for both the atomic oxygen and sulfur dioxide. The hybrid code itself does directly provide the local electron densities, this parameter is computed as the total sum of ion densities of

the torus background ($O^+$, $S^+$, $S^{++}$, $SO^+$, $SO2^+$) for the background electron density $n_{e,bg}$, and ion densities produced by electron impact and photo-ionization for the pickup electron density $n_{e,pu}$.

| Atmospheric model | | | | |
|---|---|---|---|---|
| Model | Components | | | |
| Parameter | SO$_2$ | SO | S | O |
| $n_{\mathrm{atm0}}$ [$10^8$ cm$^{-3}$]* | 2 | 0.2 | 0.1 | 0.1 |
| $H_{\mathrm{atm}}$ [$R_{Io}$] | 0.045 | 0.06 | 0.09 | 0.09 |
| $W_{\mathrm{int}}$ | 0.975 | | | |
| $W_{\mathrm{pole}}$ | 0.02 | | | |

**Table 3.** Parameters of neutral atmosphere models used in multi-species hybrid simulations. * for simulation run I0-C the parameters $n_{\mathrm{atm0}}$ are increased by a factor of 2.

The collisionless model (I0-NC and I0-NC-ND) assumes very low collisionality, so that there is no exchange of energy between the two electron (background and pick-up) populations. Therefore, the background electron population keeps its initial temperature, $T_{e,bg} = T_{e0}$ during the whole simulation and the pick-up population remains cold, $T_{e,pu} \approx 0$. The electron

excitation rate for each source then reads as $N_{\mathrm{ei}} = n_n n_{e,\mathrm{bg}} \nu_{\mathrm{ei}}(T_{e0}) \Delta V$. The collisional model (I0-C) assumes, on the contrary, high collisionality so that the background and pick-up electrons instantaneously equilibrate into a single Maxwellian population with temperature given as a weighted average of the background ($T_{e,bg} = T_{e0}$) and pick-up ($T_{e,pu} \approx 0$) electron temperatures. The total electron temperature is then $T_e = (n_{e,bg} T_{e,bg} + n_{e,pu} T_{e,pu})/n_e = n_{e,bg} T_{e0}/n_e$, and $n_e = n_{e,bg} + n_{e,pu}$ is the total local electron density. The excitation rate is hence $N_{\mathrm{ei}} = n_n n_e \nu_{\mathrm{ei}}(T_e) \Delta V$. In addition to photons produced by the thermal

electrons, we equally compute photons emitted via the energetic electron beams. These electron beams are believed to be source of the leading and secondary spots within the auroral Io footprint on Jupiter (Bonfond et al., 2008). The interaction of Io with the torus plasma within Jovian magnetosphere induces Alfvén wings structure. The traveling wave generates highly energetic electrons by still discussed mechanism, e.g., double layers or Fermi acceleration (Jacobsen et al., 2010). The electron beams propagate to both Jovian polar regions and travel across the Io's wake, where they interact with the Io's atmosphere and

produce additional source of electron excitation. Here, the electron beam density and its spatial distribution is modelled after



Roth et al. (2011), cf. equation (2), as

$$n_{e,beam} = \frac{1}{2} n_{e,beam0} \left\{ \tanh\left[3\left(x - x_{aw}(z)\right) + 1\right] + 1 \right\} \cdot \exp\left[-10\left(y - y_{aw}(z)\right)^6\right] \cdot c(x,y) \tag{4}$$

$$c(x,y) = 1 \quad \text{for} \quad (x - x_{aw})^2 + (y - y_{aw})^2 > R_{Io} \tag{5}$$

$$c(x,y) = 0.5 \quad \text{for} \quad (x - x_{aw})^2 + (y - y_{aw})^2 \leq R_{Io}, \tag{6}$$

with the average beam electron density $n_{e,beam0} = 6.4\,\text{cm}^{-3}$. The velocity distribution is assumed to be mono-energetic (delta function) with energy of $350\,\text{eV}$ according to (Williams et al., 1999; Frank and Paterson, 1999). The position of the Alfvén wings center $x_{aw}$, $y_{aw}$ is, for given position $z$, calculated from the Alfvén characteristics. Furthermore, $c(x,y)$ is a geometrical factor which reflects the fact that the beams are bi-directional and the areas hidden geometrically by the moon are thus exposed only to the beam coming from one direction.

## 3  Results

Observations of Io's auroral emission are typically provided either by ground-based and space telescopes (Roesler et al., 1999, e.g.) or by on-board cameras of different spacecraft missions (Geissler et al., 1999, 2001, 2004, e.g.) showing the air-glow as a planar projection for a given line of sight. The observations are often dominated by two key features: two bright spots close to the sub- and anti-Jovian limb concentrated around low latitudes. These equatorial spots move up and down with the ambient magnetic field of Jupiter (Retherford et al., 2000). In our model the structure of various photon emission rates is naturally equivalent. Since all the neutral atmosphere components have similar spatial distribution, we note that local rate of any typical emission observed from Io (like OI1304, OI1356, SI1479, or SI1900) would have similar structure and would scale according to magnitude of efficiency of given excitation reaction. We therefore limit our results to OI1304 emissions only. In order to derive similar results from our global hybrid simulations being comparable to the real observations, we first define three representative view directions with respect to the magnetospheric plasma flow, as it can be seen by an observer far away from Jupiter. First two panels of Figure 1 show local rate of OI1304 emissions in two representative planes as obtained from the simulation I0-NC-ND. The third panel then defines the three case viewing directions, namely the view direction (A) along the $+X$ axis (from upstream to downstream direction), direction (B) along the $+Y$ axis (looking towards Jupiter from anti-Jovian to sub-Jovian hemisphere), and finally direction (C) along the $-X$ axis (when looking into Io's wake going from downstream to upstream direction). These three views correspond to looking at Io from Earth when it is located at the dusk side of Jupiter (A), at the sub-solar point (B), and at the dawn side of Jupiter (C). The three different views are also depicted in the first two panels of Figure 1 by white arrows.

The overall auroral emission patterns are then obtained by integrating the local emission rates along the predefined line of sights for each of the three selected simulation runs. Resulting auroral patterns are displayed in Figure 2 where the individual rows are related to the simulation runs I0-C, I-NC, and I0-NC-ND respectively, and the columns from the left to right show the view directions A, B, and C. The general structure of the auroral pattern is found to be similar in all three simulations and the actual intensity is correlated with plasma/neutral densities in agreement with observations (Retherford et al., 2000). Simulations





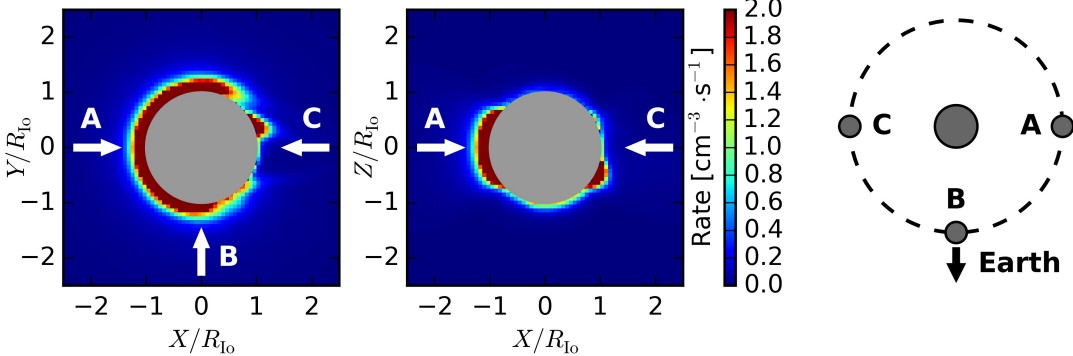

**Figure 1.** Local rate of neutral oxygen photon emissions at wavelength of 1304 Å obtained from simulation I0-NC-ND. Left panel shows results in plane $(X, Y, Z = 0)$ and middle panel shows results in plane $(X, Y = 0, Z)$. White arrows show three viewing directions on the system, (A) view in upstream-downstream direction, (B) view from anti-Jovian side of Io to sub-Jovian side, and (C) view in downstream-upstream direction. Sketch on the right shows Io at relative positions with respect to Jupiter and Earth corresponding to the three viewing directions.

produce comparable distributions of the emission rate around Io except for the wake region. In consequence integration of the rate along some direction results in the brightest auroral regions located at the sides of Io. These brightest regions are at the

flanks of Io in the upstream-downstream view and downstream-upstream view and on the upstream side for the view in anti-Jovian to sub-Jovian direction. However, the detailed morphology of the aurora clearly differs for the three simulation runs and is therefore affected by the simulations setup and model approach used.

Comparing the collisional (run I0-C, top row in Figure 2) and non-collisional approach (run I0-NC, middle row in Figure 2) there are two main differences in the derived auroral patterns, (*i*) the peak intensity in case of I0-C is lower by about a factor

of 10 compared to I0-NC, and also (*ii*) there are no significant flanks for I0-C in case of the upstream-downstream (A) and downstream-upstream view (B) (compare panels A1 and C1 wrt. panel B1 in Figure 2). In addition, in case of the collisional approach, the brightest spots are slightly lifted above the surface of the moon when compared to the non-collisional model (compare, e.g., panels B1 and B2). The auroral patterns further differ in case Io either does or does not possess the internal induced magnetic dipole, i.e., when comparing the middle and bottom row of Figure 2. This feature is evident mainly on the

sub-Jovian side of Io (see, e.g., left sides of panels A2 and A3) while the pattern on the anti-Jovian side of Io seems to be more uniform with latitude (right sides of panels A2 and A3). The auroral pattern on the sub-Jovian side is rather localized and asymmetric with respect to Io's equator so that the core of the emissions shifts below the equator in case of I0-NC (panel A2) while there is an wider spot shifted more towards the equator for the I0-NC-ND run (panel A3).

Effects of background magnetic field on the aurora's topology are known from observations which indicate that the location

of the near-equatorial spots corresponds to the location of the magnetic field tangent to the surface (Roesler et al., 1999; Retherford et al., 2000; Geissler et al., 2004). To possibly analyze and illustrate such effects we now turn to show how the sources of aurora emissions are distributed across Io's surface. Figure 3 displays the distribution of the OI1304 emission rate as

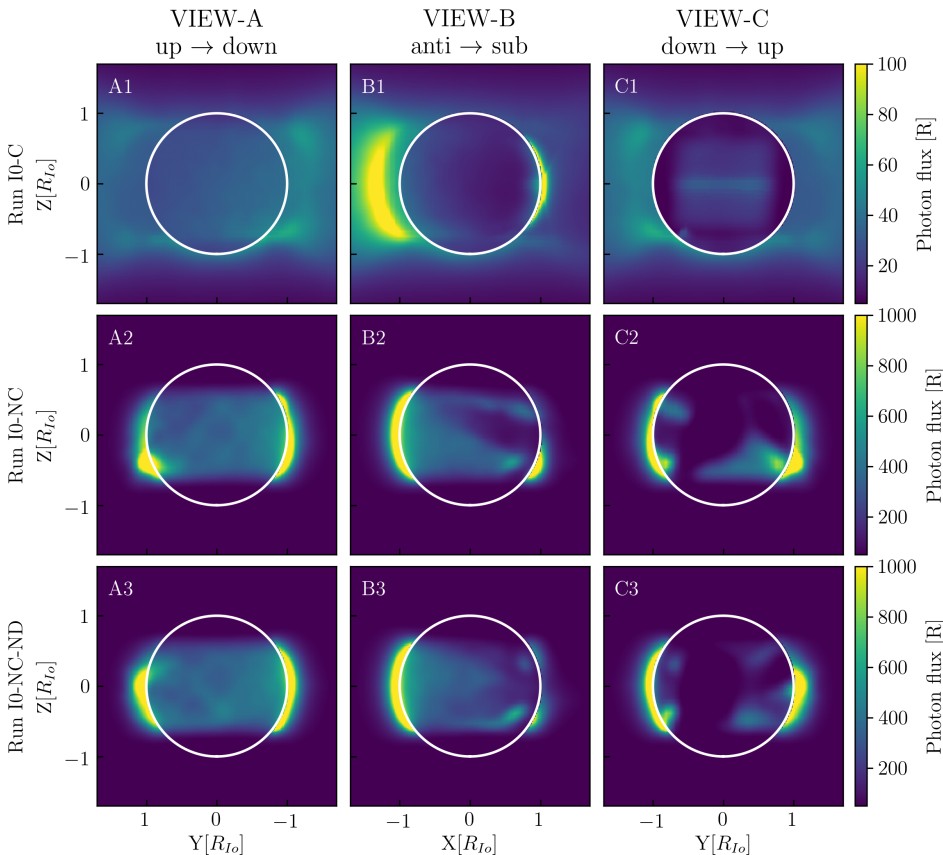

**Figure 2.** Photon flux in auroral patterns as an observer would see them from Earth's direction for three Io's positions: view A from upstream to downstream - left panels, view B towards Jupiter - middle panels, and view C from downstream to upstream - right panels. Top panels show results from the simulation I0-C, middle panels from the simulation I0-NC, and bottom panels from the simulation I0-NC-ND.

a function of longitude and latitude, again for all three selected simulation runs. The auroral maps (left column of Figure 3) are obtained by integration of the emission rate above a given position from the surface up to the height of $0.3R_{\mathrm{Io}}$. Positive latitudes correspond to the northern hemisphere of Io and the longitude is measured from the downstream point in the anti-clockwise direction (when looking on Io from above), longitudes of the sub-Jovian point (SJ), upstream point (UP) and anti-Jovian point (AJ) are marked in the figure for reference with vertical white dotted lines. In the same coordinates we plot the corresponding relative intensity of the near-surface radial magnetic field ($|B_r|/|B|$ at $0.1\ R_{Io}$ above the surface) with the magnetic equator ($B_r = 0$) for the initial conditions indicated with solid black line and magnetic equator as derived from the simulated data shown by white dashed line. The color scale used for the relative radial magnetic intensity indicates regions with quasi-tangential ($|B_r|/|B| < 0.5$) and quasi-normal ($|B_r|/|B| > 0.5$) magnetic field lines with red and blue shades, respectively.

A common feature of auroral distributions in case of non-collisional simulations (I0-NC and I0-NC-ND) is that most of the emissions are produced in the upstream hemisphere (longitudes $90° - 270°$) and that the distribution on this hemisphere





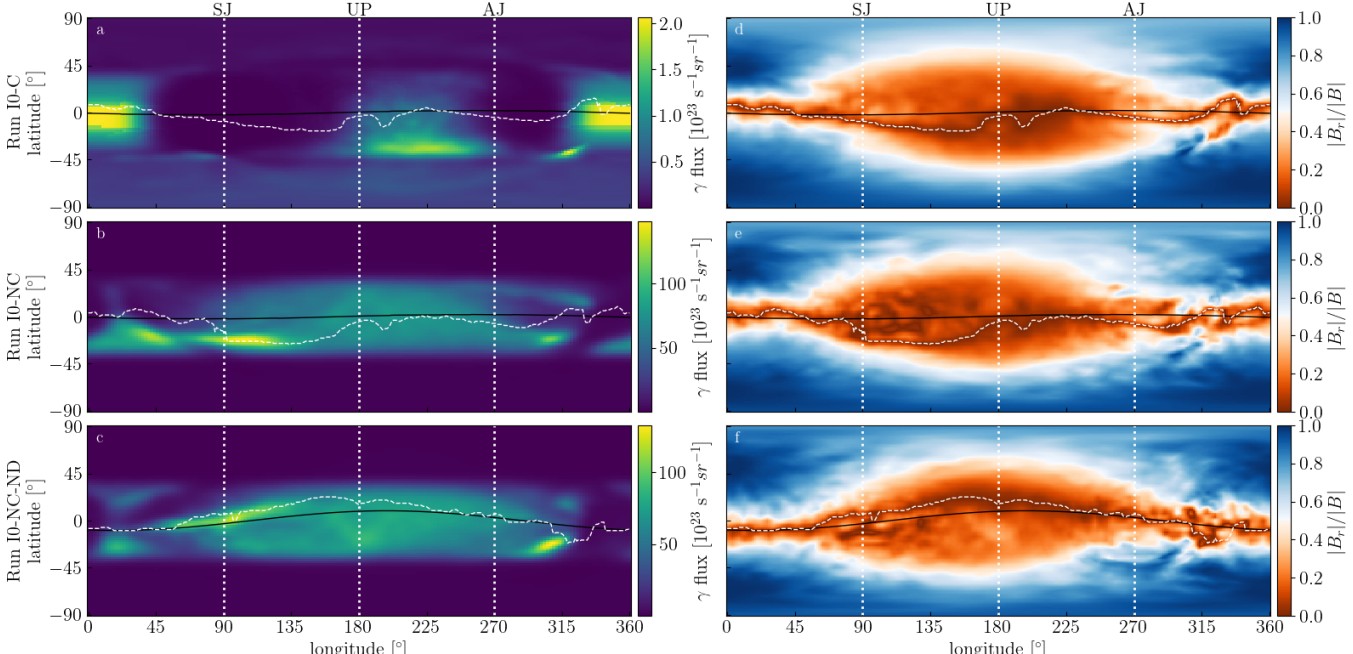

**Figure 3.** Distribution of the photon flux (left panels) from the OI1304 emission on the surface of Io from the three simulation discussed: I0-C - top panels, I0-NC - middle panels, I0-NC-ND bottom panels. The right panels show the map of the radial component of the local magnetic field at a distance $1.1R_{Io}$ from the Io's center. The color distinguishes here quasi-tangent (red) from quasi-radial (blue) magnetic field. The black solid line indicates theoretical magnetic equator given by the initial conditions of the simulation setup (background field and induced dipole - if taken into account) without local plasma perturbation. The white dashed line indicates magnetic equator (a minimum of the radial component - area of quasi tangent magnetic field) respecting local conditions and magnetic field topology. Vertical white dotted lines mark longitudes of the sub-Jovian point (SJ), the upstream point (UP), and the anti-Jovian point (AJ).

is rather uniform (with limited validity for I0-C), in agreement with the patterns shown in Figure 2. On the other hand the

downstream hemisphere exhibits more discrete auroral features which are locally brighter than the emissions from the upstream side. Moreover it can be observed that the downstream emissions for I0-NC and I0-NC-ND split into two bands located at latitudes $\vartheta \approx \pm 30°$ with higher intensity in the southern hemisphere. Typical longitudes for the bright downstream features are $\varphi \approx 25° - 50°$ for the sub-Jovian side and $\varphi \approx 270° - 300°$ on the anti-Jovian side. The auroral distribution again differs for the collisional simulation run I0-C. Here the pattern in the upstream hemisphere is rather asymmetric with peak intensities

south below the equator at longitudes $\varphi \approx 180° - 270°$. On the other hand in the downstream hemisphere the auroral emissions are symmetric around downstream point $(0°,0°)$ in a narrow band extending $\approx \pm 25°$ in latitudes and $\approx \pm 30°$ in longitudes.

As discussed above, Io's background magnetic field and as well as the induced dipole field has a significant impact on the interaction structure. This impact already observed in Figure 2 is again found for simulation runs I0-NC and I0-NC-ND in Figure 3, now with direct comparison to the local magnetic field topology. The effect of the induced dipole on the magnetic

field structure, and namely location of the tangent points, is demonstrated when comparing the panels $e$ and $f$ of Figure 3. In





case of the simulations run I0-NC-ND without the induced dipole (panel $f$) the simulated magnetic equator (white dashed line) is pushed into the northern hemisphere in the upstream region and fairly well follows the initial simulation conditions (solid black line). On the contrary, when including the induced dipole in the simulation run I0-NC, the simulated magnetic equator, affected by the kinetic plasma processes, is now pushed more to the southern hemisphere in the upstream region and deviates

from the initial conditions namely for $\varphi \approx 80° - 180°$. As for I0-NC, a qualitatively same structure is found for the simulation run I0-C, however, with slightly reduced deviations from the initial conditions. The trace of the magnetic equator is equally over-plotted on the auroral patterns in panels $a$, $b$, and $c$ of Figure 3. For the non-collisional simulation runs (panels $b$ and $c$) we see the bright auroral features around the sub-Jovian point to be well tracked by the tangential magnetic field lines. Such correlation is not evident for the simulation run I0-C (panel $a$). Also the bright auroral spot observed for all three simulations

runs in the southern hemisphere around $\varphi \approx 300°$ is apparently not directly linked to the local tangent field lines but rather corresponds the a localized region of an almost radial magnetic field.

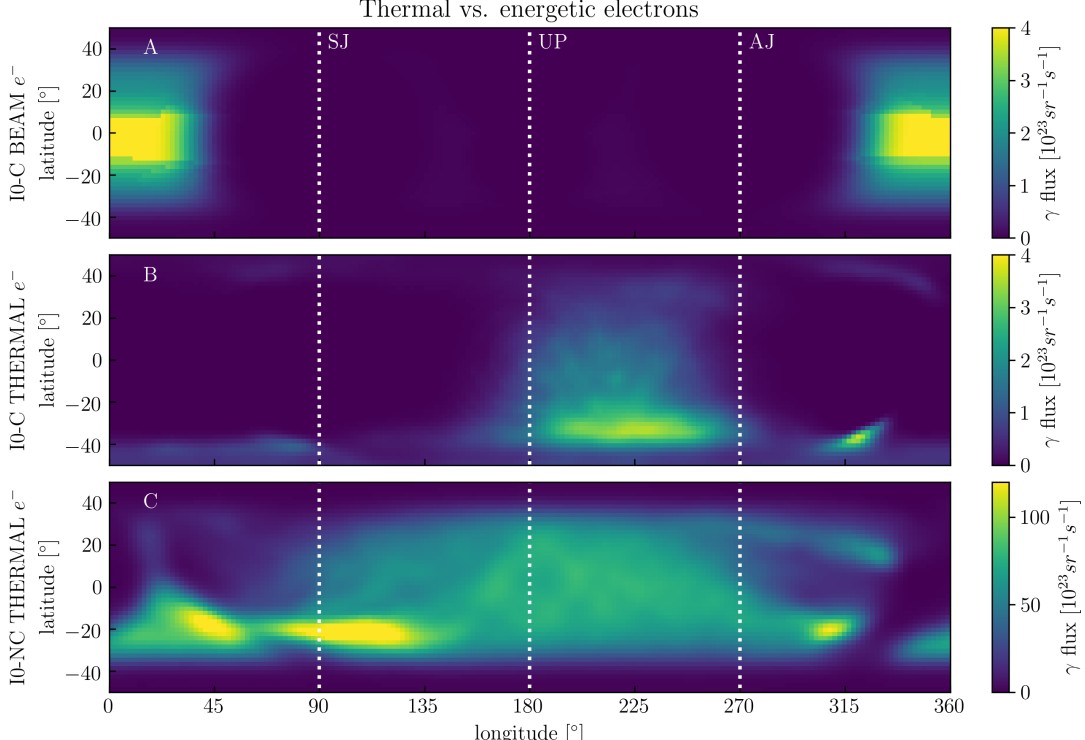

**Figure 4.** Comparison of various contributions to the emissions shown in the longitude latitude maps. The top panel shows emission rate (in terms of photon flux) of the OI1304 caused by the electron beams, middle panel shows the emissions caused by the thermal electrons in the collisional model, and the bottom panel shows the emissions caused by the thermal electrons in the collisionless mode.

The auroral results presented so far dealt with OI1304 emissions produced by both the population of thermal electrons and the energetic electron beams. In Figure 4 we demonstrate how these two individual components contribute to the overall



auroral patterns. We map again the auroral intensities as a function of the local longitude/latitude but now the top panel $A$ of
Figure 4 shows the photon flux for simulation I0-C as created by the energetic electrons only while the middle panel $B$ plot
photon fluxes emitted by the thermal electrons (again for run I0-C). The figure is completed with the bottom panel $C$ showing
the effect of thermal electrons in the simulation run I0-NC. Note that the auroral pattern from energetic beams is identical in
structure for all simulation runs (as it is given by equation (6) being dependent on the initial plasma conditions only), and the
auroral intensities will just decrease by a factor of 2 for simulations I0-NC and I0-NC-ND due to lower density of the neutral
atmosphere.

The beam auroral pattern (panel $A$), as given by both the model of the neutral atmosphere (1) and the model of electron beam
density (6), has its peak intensities at low latitudes in the downstream region. The beam and thermal auroral peak intensities
are comparable in case of the collisional simulation run I0-C (compare panels $A$ and $B$) and thus the electron beams provide a
substantial contribution to the overall auroral pattern. The beam contribution is well recognized in the downstream hemisphere
in the panel $a$ of the Figure 3 and can be observed equally in panel $B1$ of Figure 2 as the thin near-surface flank on the right side
of the moon. In the case of the non-collisional simulations (compare panels $A$ and $C$), the beam auroral intensity is significantly
lower with respect to those produced by the thermal electrons so that the beam contribution is not visible in the overall auroral
structures (see corresponding panels in Figure 3). It has to be also stressed out that the beam auroral pattern has no correlation
to the magnetic field topology in our results as this is not included in the beam density model (6).

**4  Discussion**

The virtual telescope pictures as presented in Figure 2, showing auroral morphology of Io as it would be observed from three
main directions, are to certain extent in fair agreement with the real observations (Roesler et al., 1999; Geissler et al., 1999,
2001, 2004, cf.). The collection of local emission rate along the view direction naturally produces bright spots at the flanks
of Io where the line of sight goes through longest region of emission. The characteristic bright spot is well reproduced on
the sub-Jovian side which is, however, not the case of the anti-Jovian side where the auroral pattern is extended into a wider
band around the equatorial plane. Comparing the upstream and downstream flanks we see significantly higher intensity at
the upstream hemisphere. The correlation between the real and virtual auroral patterns is much less evident in case of the
collisional model which favours the non-collisional approach in agreement with Šebek et al. (2019). In the collisional model
the background electrons are assumed to be in thermal equilibrium with the initially cold electrons from ionization processes.
In regions with the most intense ionization, i.e., close to the surface of the moon, the local electron temperature is significantly
reduced and so are therefore lowered auroral emissions (as the effective cross-sections of the excitation reactions naturally
rapidly decrease for cold electrons). This has two direct consequences, both evident in the upper row of Figure 2: the most
intense emissions are lifted to high altitudes, $\approx 0.2\ R_{Io}$ above the surface of the moon, and the overall intensity is significantly
reduced since the density of the neutral model atmosphere quickly falls with the radial distance. On the other hand the bright
auroral spots in the case of the non-collisional approach are bound to the surface where both the neutral and electron densities
reach their maxima.





Observations of Io's auroral emissions are typically dominated by emissions around the sub and anti-Jovian flanks, mostly as in a form of bright spots near the Io's equator (Roesler et al., 1999; Geissler et al., 1999). The existence of the spots assumes diversion of incoming electron streams being concentrated in regions of tangential magnetic field lines, not directed towards or away of the moon's surface (Saur et al., 2000). The location of the spots is therefore believed to be driven by the configuration of the magnetic field, namely the by the position of the magnetic equator, as it is often indeed found in many observations. Analysis of Io's ultraviolet aurorae indicates that the brightest equatorial spots in average diverted from sub-Jovian and anti-Jovian meridian by $\approx 10° - 30°$ (Retherford et al., 2000). Under this assumption Roth et al. (2017) argue that a setup consistent with observations of Io's aurora is that either Io doesn't have an induced magnetic field or the induction takes place in its highly conductive core. However, other observations exist with the spots being largely scattered far of the expected locations (e.g. Retherford et al., 2000).

We presented the distribution of the radial component of the magnetic field with respect to Io's surface at a distance $0.1R_{\mathrm{Io}}$ above the surface (see the right column of Figure 3). The magnetic field is approximately tangential to the surface at locations with small radial component. Each plot shows also the latitude of the tangent magnetic field for the initial magnetic field in a given simulation (solid black line), that is for the magnetic field topology without perturbation from plasma interaction. The simulated magnetic equator (white dashed line) is clearly distorted from its initial shape by many acting plasma processes. Our results exhibit similar features to those obtained recently by Roth et al. (2017). Namely there is a broad range of latitudes with almost tangential magnetic field on the upstream hemispheres (red color scale) while on the downstream hemisphere the region of tangential magnetic field is more limited towards the lower latitudes.

We find a good agreement of the bright emissions aligned to the local magnetic equator for the non-collisional electron model in the sub-Jovian hemisphere. For the initial configuration without the induced magnetic dipole the bright emissions are here located at rather low latitudes while inclusion of the induced dipole moves the spots to $\varphi \approx -30°$ being in better agreement with Retherford et al. (2000), giving priority again to the I0-NC configuration in agreement with Šebek et al. (2019). Roth et al. (2017) also noted that if an induced field was not present at Io, an alternative explanation for magnetic field perturbations could be found by including asymmetries in Io's atmosphere. However Šebek et al. (2019) achieved good fits to Galileo data from all flybys while assuming a simple prescription of the atmosphere. Moreover, their results obtained with longitudinally asymmetric atmosphere models don't exhibit significant difference in the magnetic field profiles when compared to the symmetric atmosphere case. This suggests that the presence of an induced dipole field at Io is the more probable explanation for the Galileo magnetic field measurements. We stress out that the correlation between magnetic field direction and inferred morphology of Io's aurora obtained from our simulations is not as distinct as in observations in the case of the anti-Jovian longitudes. The emissions are rather broad in space around the anti-Jovian point and lower in the brightness. A well localized spot exists at $\varphi \approx 300°$, however, this location rather corresponds to almost radial field lines. Even weaker correlation is found for the collisional approach. Here the reason may be that the emissions take place at a wide range of higher altitudes where the magnetic field topology can differ with radial distance compared to the structure given in panel $d$ of Figure 3.





In addition to the thermal plasma, energetic bi-directional electrons were observed in Io's vicinity (Williams et al., 1999, e.g.) with energies ranging from keV to hundreds of keV (Frank and Paterson, 1999, 2002)). In the present approach we assume the lower energy beams of $\approx 350\,eV$ with energy flux of about $4\,erg\,cm^{-2}\,s^{-1}$ based on Frank and Paterson (1999). The beams are expected to be relatively narrow (in $Y$ direction) extending far into the wake, however, in the far wake the neutral density

vanishes, and no photon emissions can take place anyway. Moreover, inside the Alfvén wings half of the energy flux is shielded by the moon's body itself. The theoretical contribution of the energetic electrons is presented in Figure 4. Beam emissions are by the model constrained to the downstream hemisphere where the concentration of thermal electrons is low. Still the intensity of the auroral emissions due to the energetic beams is found much lower compared to the thermal component so that their role in the overall aurora's morphology can be neglected (except for the case of the collisional model which has been, however,

mostly rejected due other non conformances).

## 5    Conclusions

In this paper, we have analyzed the aurora morphology of the Jupiter's moon Io, depending on the different configurations of the magnetospheric background. Recent studies of Io's aurora observations mainly employ numerical models based on the MHD approach (see, e.g., Roth et al. (2017), and references therein). For our study we have used global numerical simulations

of the Io's interaction with the magnetospheric plasma based on the hybrid code as developed by Šebek et al. (2019). This model self-consistently resolves the plasma processes down to the ion kinetic scales and uses multiple-species approach to implement the chemistry of both the background plasma torus and the Io's neutral atmosphere with their mutual ionization processes. We have selected three simulation setups in order to examine the effect of (*i*) the local electron temperatures, and (*ii*) of the magnetic field configuration. The model of Io's aurora emissions for the representative case of OI1304 were then

computed from the simulated plasma and magnetic field parameters using empirical cross-sections of the effective reactions.

First, the derived auroral emissions were presented in the form of virtual telescope images. With this technique we have analyzed the effect of the observer's position with respect to the torus plasma flow on the resulting aurora's patterns. Second, we created a set of topographic maps to compare the distribution of the auroral emissions on the Io's surface with the topology of the local magnetic field lines. Our simulated aurora exhibits several similar features compared to real Io's observations. The

integrated photon flux is highest on the flanks of Io and on its upstream side. These features are caused to a large extent by the aspects of viewing geometry. The surface distribution of auroral emissions is uniform on the upstream side of Io and exhibits discrete localized and relatively brighter features, namely at sub and anti-Jovian side of Io.

In the simulations setups we have used two methods for computation of the local electron temperature. The, so called, collisional approach implements thermal equalization of the background and pick-up electrons which effectively decreases the

local electron temperature in the Io's atmosphere. On the other hand the non-collisional method assumes no thermal interaction of the latter two electron populations thus only the thermal background electrons produce auroral emissions and not the pick-up population. The auroral images as produced from the non-collisional approach were found to be reasonably closer to real observations. Similar findings were published by Šebek et al. (2019) when comparing the simulated Io's flybys with real

measurements from the Galileo mission. Moreover, the non-collisional set-up revealed strong correlation of the bright auroral
spots near the sub-Jovian point with the regions of a nearly tangential magnetic field topology. Two different setups of the
non-collisional approach were examined with respect to initialization of the background magnetic field: with and without an
Io's induced magnetic dipole. The position of these spots was found closer to real observations (Retherford et al., 2000) when
the induced dipole was present, in agreement with the recent hypotheses and again similar to other comparisons with Galileo
data as published by Šebek et al. (2019). A rather poor agreement with the real aurora observations was found for modelled
patterns on the anti-Jovian side, also with less evident correlation to the magnetic field topology, which is a main subject for
the next research work.

Additional analysis was performed to quantify a potential contribution of energetic electron beams (Williams et al., 1999).
The energetic beams are observed mainly in region of the Io's wake (downstream hemisphere) with energies up to hundreds of
keV. They are naturally capable to contribute to the auroral emissions, however their densities are significantly smaller when
compared to the thermal electron background. Indeed we found the contribution of the beams in the non-collisional approach
to be negligible for the overall auroral pattern structure. The auroral spots as produced by the energetic beams were significant
only in case of the collisional approach, however, this model did not conform to the real observations. Our model thus predicts
the beam emissions to be observable in the downstream region in case of significantly enhanced energy fluxes.

In general the Io's auroral model based on the global hybrid numerical simulations was proven as a useful tool to study the
Io's aurora patterns. In turn we have also demonstrated that Io's aurora observations can provide indirect methods how to study
many plasma processes in the Io's interaction region.

*Code and data availability.* Presented data and corresponding processing scripts are available at http://doi.org/10.5281/zenodo.4472602

*Author contributions.* The global hybrid models of Io, being the corner stone of the presented analysis, and initial ideas towards modeling
the Io's auroral patters were founded by OŠ as a part of his PhD study under supervision of PT. An extended comparative analysis of the
auroral patterns for representative simulations runs was consequently conducted by ŠŠ. DH mostly contributed to numerical processing and
visualization of the simulation data sets. All authors have equally contributed to the writing and preparation of the manuscript.

*Competing interests.* The authors declare that no competing interests are present.

*Acknowledgements.* The present research was conducted by support of Czech Science Foundation under project n. 17-08857S. All numerical
simulations used in this study were performed using the parallel HPC facility Amálka running at the Institute of Atmospheric Physics CAS,
thanks to last major system upgrade achieved via EU funding support from project CZ.2.16/3.1.00/24512.





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
