# Peer review of "Io's auroral emissions via global hybrid plasma simulations"

_Annales Geophysicae, 2021_

## Referee Comment (RC1)

Referee Report on "Io's auroral emissions via global hybrid simulations" by Stepan Stverak et al.

This manuscript presents hybrid (kinetic ions, fluid electrons) modeling results of the interaction between Io and Jupiter's magnetosphere. The calculated plasma quantities are applied to determine spatially resolved maps of the auroral emission intensity for different representations of the moon's electromagnetic environment and the electron-neutral interactions within its neutral gas envelope.

The manuscript is well organized and (mostly) well written. There are a few recurring grammar errors ("the Io's", "the Jupiter's", etc.) which could be eliminated through careful proofreading.

However, I do have several major concerns on the applied methodology and the interpretation of the results, as detailed below. Convincing rebuttal is required before this manuscript can be considered for publication. I therefore recommend to return this manuscript to the authors for major revisions.

Detailed comments:

1) In several passages, the authors claim that adequate modeling of Io's magnetospheric interaction requires the inclusion of an intrinsic (induced) dipole moment at the moon. However, their study does not take into account the results of Aljona Bloecker et al. (2018, https://agupubs.onlinelibrary.wiley.com/doi/full/10.1029/2018JA025747). This paper demonstrates that the magnetic field perturbations observed at Io can, in large portions, be explained through the plasma interaction with a *longitudinally* asymmetric atmosphere. The presence of an induced dipole is not necessary to reproduce the magnetic signatures observed during the Galileo flybys. In this sense, the results of the Dols et al. (2012) paper on the potential presence of an induced field (referred to in the manuscript) have become somewhat outdated. The longitudinal atmospheric inhomogeneities identified by Bloecker et al. (2018) give rise to regions of enhanced current density within Io's main Alfven wings (referred to as "Alfven winglets" by these authors). While the magnetic field signatures of an induced dipole and the Alfven winglets may look similar when sampled along a spacecraft trajectory, the overall shape of the field lines in both setups is very different. To obtain adequate constraints on the auroral emission morphology, I strongly suggest that the authors present additional simulation results without an induced dipole, but with an adequate representation of the longitudinal atmospheric asymmetries described by Bloecker and co-workers. In my opinion, this is a very important point that deserves far more attention than a few lines of additional text.

2) The authors claim that a major strength of their approach is the description of ion kinetic effects in the plasma flow near Io (see, e.g., section 1 and lines 233-234). However, based on the upstream parameters from table 2, the ion gyroradii in the plasma near Io are, at most, on the order of only 5.2 km (oxygen and doubly-charged sulfur), 10.3 km (singly-charged sulfur), and 20.7 km (sulfur dioxide). Not only are these gyroradii tiny compared to the size of Io, but I do not think that the authors' hybrid model (or, more general, any three-dimensional hybrid model!) would be able to resolve gyration of the oxygen and sulfur ions anywhere near Io, nor the gyration of the sulfur dioxide ions in the stagnated and strongly magnetized plasma near the moon. Section 2.6 of the *Sebek et al. (2019)* paper gives a resolution on the order of 13 km for all three model setups considered here, and I would say that this is too coarse to resolve any ion kinetic effects in the vicinity of Io where the plasma bulk speed is way smaller than the upstream value and the plasma is cold. In particular, I am concerned that the model does not adequately resolve the ionospheric Hall effect, which was found crucial to explain the observed asymmetries in the magnetic field and the flow deflection pattern observed near Io by the Galileo spacecraft (see *Bloecker et al., JGR, 2018,* but especially *Saur et al., JGR, 1999,* https://agupubs.onlinelibrary.wiley.com/doi/abs/10.1029/1999JA900304). It would be very important to discuss how the ionospheric Hall effect maps into the auroral emissions and to what degree the

corresponding auroral signatures are resolved by the model presented here. In general, the authors should explain what kinetic effects are actually resolved in their model and how these effects impact the results. It would also be important to know if these effects generate any novel features in the modeled aurora morphology, compared to the results of, e.g., the two-fluid approach of *Roth et al. (2011)*. To be clear, I do not blame the authors for not reaching the resolution required to resolve ion kinetic effects (the "culprit" is the strong magnetic field near Io): this may not even be feasible with current computing resources (see also my next point). However, this shortcoming raises the question of why the application of a hybrid model is even necessary at Io and what new features of the aurora can be revealed by such an approach, compared to earlier (less complex) models of the interaction.

3) Also, skimming through the Sebek et al. (2019) paper (upon which the present manuscript is based), I noticed that the authors *do not* study Io's actual plasma interaction, but the radius of the obstacle is artificially reduced by a factor of 10 (!), mainly for reasons of computational efficiency. Since all the hybrid simulation results used in this manuscript have been taken from the Sebek et al. paper, I assume that the present study also considers a "downscaled" Io interaction scenario. This important simplification is not even mentioned anywhere in the manuscript. The authors need to explain in detail why they believe that their approach is suitable to produce realistic auroral emission maps (especially when it comes to the variation of the emission intensity with altitude), how the "downscaling" of Io's atmosphere will affect the results and the merit of any comparisons to telescope observations.

4) I do not understand the method described in line 148 to calculate the electron temperature. It seems that the authors discriminate between different electron populations in the plasma (bg /"background" and pu/"pick-up"), assuming that these electron densities are identical to the corresponding ion densities. This implies that, in the proposed model, each ion population is "accompanied" by its own electron population. However, in reality there is only ONE electron population, and it is impossible to assign a certain electron to a specific ion population. While such an assumption may or may not be justified within the framework of this model, it seems to me that this is a misrepresentation of the concept of quasi-neutrality. Only a single electron density enters the quasi-neutrality condition and the governing equations of the hybrid model, and it is not clear to me how the evolution of the quantities $n\_bg$ and $n\_pu$ is treated in the model presented here. Several MHD models solve a separate continuity equation for the ionospheric electrons (e.g., the Europa code of Bloecker et al. (2016), see their equation 10). However, this is usually not done in the hybrid approach. In my opinion, far more discussion of this simplification and its consequences for the validity of the results needs to be included.

5) As already briefly stated above, the manuscript should discuss in detail how/why the applied methodology and the results of this modeling effort go significantly beyond those already presented by Lorenz Roth et al. (2011) and other preceding modeling studies (from, e.g., Joachim Saur et al.). Compared to preceding studies by other groups, what are the novel and innovative contributions of the results presented here? Also, latest observational results from Roth et al. (https://www.sciencedirect.com/science/article/pii/S0019103520303018) may be relevant in the context of this study (the referee is NOT Lorenz Roth).

6) Around line 160: How do the authors calculate the center of the Alfven wings? If an induced dipole is taken into account, the Alfven wing tubes are shrunk (compared to the cross-section of the obstacle) and

are also displaced with respect to the moon's disk. I recommend that the authors have a look at the paper of Neubauer (1999, https://agupubs.onlinelibrary.wiley.com/doi/abs/10.1029/1999JA900217) and subsequent observational evidence from Volwerk et al. (2007) at Europa (https://angeo.copernicus.org/articles/25/905/2007/).

7) Overall, it would be greatly appreciated if the authors provided a stronger connection between their modeled auroral emissions and the three-dimensional structure of the plasma interaction signatures near Io. I did not learn much about the physics from this study, beyond the already known importance of regions where the magnetic field is tangential/perpendicular to the surface. Based on the available results, the authors could build a much more compelling case by investigating in depth the physical mechanisms that lead to their modeled auroral patterns. For instance, how do the shrunk/displaced Alfven wing tubes map into the aurora, what is the role of magnetic pile-up at Io's ramside, does the tilt of the Alfven wings have any discernible influence on the emission maps, and what is the role of local atmospheric inhomogeneities (like those observed near Tvashtar)? The authors' hybrid model even allows to calculate three-dimensional maps (or corresponding 2D cuts) of the electron density and bulk velocity ($u\_i$-curl $B/mu\_0$, where $u\_i$ is the ion velocity) which would be extremely helpful in elucidating the connection between Io's auroral emissions and processes in the moon's plasma environment. More detailed discussions of the physical mechanisms, exploiting the full potential of the model, would greatly strengthen this paper.

8) To validate the model, it would be very helpful if authors showed a side-by-side comparison between their modeled auroral patterns and actual observations. Throughout the paper, the authors claim that various features of their modeled aurora are in agreement with observations, but the manuscript does not show an actual comparison to substantiate this claim. It is fully acknowledged (and evident to any knowledgeable reader) that such a comparison cannot quantitatively reproduce all the observed features of Io's aurora, but a side-by-side comparison would make it very straightforward to identify strengths and weaknesses of the model.

9) A minor issue (page 4, lines 107-108): The induced dipole moment is not antiparallel to the background magnetic field. It is antiparallel only to the horizontal (Bx, By,0) component of the field, since the north-south component $B\_z$ is nearly constant in time.